# Randomized Phase 2 Clinical Trial of Olaratumab in Combination with Gemcitabine and Docetaxel in Advanced Soft Tissue Sarcomas

**DOI:** 10.3390/cancers15194871

**Published:** 2023-10-06

**Authors:** Steven Attia, Victor Villalobos, Nadia Hindi, Andrew J. Wagner, Bartosz Chmielowski, Gerard J. Oakley, Patrick M. Peterson, Matteo Ceccarelli, Robin L. Jones, Mark A. Dickson

**Affiliations:** 1Mayo Clinic, Jacksonville, FL 32224, USA; 2School of Medicine, University of Colorado Denver, Aurora, CO 80045, USA; 3Hospital Universitario Virgen del Rocío, 41013 Sevilla, Spain; 4Fundación Jiménez Díaz University Hospital, 28040 Madrid, Spain; 5Hospital General de Villalba, 28400 Madrid, Spain; 6Center for Sarcoma and Bone Oncology, Dana-Farber Cancer Institute, Boston, MA 02215, USA; 7Harvard Medical School, Boston, MA 02115, USA; 8Jonsson Comprehensive Cancer Center, University of California Los Angeles, Los Angeles, CA 90024, USA; bchmielowski@mednet.ucla.edu; 9Eli Lilly and Company, Indianapolis, IN 46255, USAceccarelli_matteo@lilly.com (M.C.); 10Institute of Cancer Research, Royal Marsden Hospital, Fulham Road, London SW3 6JJ, UK; 11Memorial Sloan Kettering Cancer Center, New York, NY 10065, USA; 12Weill Cornell Medical College, New York, NY 10065, USA

**Keywords:** soft tissue sarcomas, olaratumab, gemcitabine, docetaxel, overall survival, progression-free survival, leiomyosarcoma

## Abstract

**Simple Summary:**

Soft tissue sarcomas (STSs) are rare and highly heterogeneous tumors that are difficult to treat. Gemcitabine plus docetaxel is an effective treatment for advanced STS. However, the prognosis for patients remains poor, and thus there is an urgent medical need for novel and effective therapies to improve long-term outcomes. The aim of the ANNOUNCE 2 trial was to explore the addition of olaratumab (O) to gemcitabine (G) and docetaxel (D) for advanced STS. Patients were randomized 1:1 from two cohorts (O-naïve and O-pretreated) to 21-day cycles of olaratumab, gemcitabine, and docetaxel. A total of 167 and 89 patients were enrolled in the O-naïve and O-pretreated cohorts, respectively. There was no statistically significant difference in the primary endpoint of overall survival between the two arms in the O-naïve population. No new safety signals were observed.

**Abstract:**

Gemcitabine plus docetaxel is an effective treatment regimen for advanced soft tissue sarcomas (STSs). However, the prognosis for patients remains poor, and thus there is an urgent medical need for novel and effective therapies to improve long-term outcomes. The aim of the ANNOUNCE 2 trial was to explore the addition of olaratumab (O) to gemcitabine (G) and docetaxel (D) for advanced STS. Adults with unresectable locally advanced/metastatic STS, ≤2 prior lines of systemic therapy, and ECOG PS 0–1 were eligible. In Phase 2, patients were randomized 1:1 from two cohorts (O-naïve and O-pretreated) to 21-day cycles of olaratumab (20 mg/kg Cycle 1 and 15 mg/kg other cycles, Days 1 and 8), gemcitabine (900 mg/m^2^, Days 1 and 8), and docetaxel (75 mg/m^2^, Day 8). The primary objective was overall survival (OS) in the O-naïve population (α level = 0.20). Secondary endpoints included OS (O-pretreated), other efficacy parameters, patient-reported outcomes, safety, pharmacokinetics, and immunogenicity. A total of 167 and 89 patients were enrolled in the O-naïve and O-pretreated cohorts, respectively. Baseline patient characteristics were well balanced. No statistically significant difference in OS was observed between the investigational vs. control arm for either cohort (O-naïve cohort: HR = 0.95 (95% CI: 0.64−1.40), *p* = 0.78, median OS, 16.8 vs. 18.0 months; O-pretreated cohort: HR = 0.67 (95% CI: 0.39−1.16), *p* = 0.15, median OS 19.8 vs. 17.3 months). Safety was manageable across treatment arms. There was no statistically significant difference in the primary endpoint of OS between the two arms in the O-naïve population, and therefore based on hierarchical evaluation no other outcomes in this study can be considered statistically significant. No new safety signals were observed.

## 1. Introduction

Soft tissue sarcomas (STSs) are tumors that develop in connective tissue from embryonically derived mesenchymal cells [1,2,3]. Although STSs are rare, accounting for ~1% of all adult cancers, they are highly heterogeneous, making them difficult to treat [4]. For ˃40 years, doxorubicin-based therapy has been a mainstay first-line (1L) systemic treatment in this setting [5,6].

Although drug combinations with doxorubicin have been investigated, including doxorubicin plus ifosfamide [7], palifosfamide [8], or evofosfamide [9], none of these trials have shown a statistically significant difference in overall survival (OS) compared to doxorubicin alone. To add to the lack of therapeutic options for STS, doxorubicin-based chemotherapy is generally avoided in older patients and those with current or previous abnormal cardiac function [10,11,12]. As a result, other effective chemotherapeutic combinations or novel therapies are required in the management of STS. 

The combination of gemcitabine and docetaxel has been shown to have activity in STS and is generally accepted as a standard 1L or second-line (2L) treatment regimen for patients following or otherwise not suitable for treatment with doxorubicin [10,13,14]. However, the prognosis for patients remains poor, and there is an urgent medical need for novel and effective therapies to improve long-term outcomes [3].

Dysfunction of the PDGF-PDGFR-α signaling pathway has been observed in STS, resulting in uncontrolled tumor growth and proliferation [15]. Olaratumab, a recombinant human immunoglobulin G subclass 1-type monoclonal antibody, binds with high affinity to PDGFR-α and blocks binding of platelet-derived factor-AA, -BB, and -CC to the receptor [16]. Preclinical studies showed a higher tumor growth inhibition in a patient-derived xenograft model of human liposarcoma with the addition of olaratumab to gemcitabine plus docetaxel (O+G+D) [17].

Here, we present efficacy and safety findings from the randomized Phase 2 portion of the ANNOUNCE 2 Phase 1b/2 trial of O+G+D in patients with advanced STS from two cohorts: O-naïve and O-pretreated. The study was initiated concurrently with a Phase 3 confirmatory study of olaratumab and doxorubicin (ANNOUNCE). While the ANNOUNCE trial was negative [18], no new safety risks were observed, prompting the decision to continue the on-going ANNOUNCE 2 trial evaluating the potential benefit of olaratumab added to a different therapeutic backbone.

## 2. Methods

### 2.1. Study Design

ANNOUNCE 2 (NCT02659020) was a multicenter Phase 1b/2 study evaluating efficacy, safety, patient-reported outcomes (PROs), and pharmacokinetic (PK) profile of O+G+D in patients with locally advanced/metastatic STS. The Phase 1b part of the study consisted of an open-label, single-arm dose-escalation assessment of safety for the combination. Phase 1b dose-escalation results were previously reported [19]. Phase 2 consisted of a randomized (1:1), double-blinded, placebo-controlled study enrolled from two cohorts: O-naïve and O-pretreated. Patients from each cohort were randomized into two study treatment arms, the investigational arm (O+G+D) and control arm (PBO+G+D).

The study protocol was approved by institutional review boards and ethics committees before initiation and conducted in accordance with the Declaration of Helsinki. All patients provided written informed consent before participation.

### 2.2. Patient Population

Key eligibility criteria included age ≥ 16 years; histologically confirmed diagnosis of locally advanced/metastatic STS not amenable to curative treatment with surgery/radiotherapy; no prior treatment with gemcitabine/docetaxel; no more than two prior lines of systemic therapies for locally advanced/metastatic disease; Eastern Cooperative Oncology Group (ECOG) performance status (PS) of 0–1; and evaluable disease as defined by Response Evaluation Criteria in Solid Tumors (RECIST 1.1).

Key exclusion criteria included diagnosis of gastrointestinal stromal tumor or Kaposi sarcoma and active central nervous system or leptomeningeal metastasis. Patients previously enrolled in other blinded studies with olaratumab were not eligible to participate in this trial.

### 2.3. Trial Design and Interventions

Patients received 21-day cycles of olaratumab (20 mg/kg (IV) Cycle 1 and 15 mg/kg (IV) other cycles on Days 1 and 8), gemcitabine (900 mg/m^2^ (IV) on Days 1 and 8), and docetaxel (75 mg/m^2^ (IV) on Day 8) based on the results of the Phase 1b dose-escalation assessment previously reported [19].

In the Phase 2 part, patients were stratified based on five factors: prior treatment with olaratumab (yes/no), number of prior systemic therapies for locally advanced/metastatic disease (0/≥1), histological tumor type (leiomyosarcoma (LMS/non-LMS)), ECOG PS (0/1), and prior pelvic radiation (yes/no).

Treatment continued until there was evidence of disease progression, death, intolerable toxicity, or other withdrawal criteria were met. If olaratumab/placebo had to be discontinued, patients could continue to receive gemcitabine and docetaxel. Study completion occurred following the primary analysis of OS.

### 2.4. Study Endpoints and Assessments

The primary objective of the Phase 2 part was OS (time from date of randomization to date of death from any cause) in O-naïve patients (O+G+D vs. PBO+G+D). OS was censored for analysis on the last date the patient was known to be alive. A key secondary endpoint was OS in O-pretreated patients. Other secondary endpoints included progression-free survival (PFS, determined by investigator assessment according to RECIST 1.1), objective response rate (ORR, defined as the proportion of patients achieving best overall response (BOR) of complete response (CR)+partial response (PR)), disease control rate (DCR, defined as the proportion of patients achieving BOR of CR+PR+SD), duration of response (DoR, defined from date of first documented CR/PR to first date of radiologic disease progression or death due to any cause, whichever was earliest), PROs, and to assess safety and PK profile of O+G+D. Safety endpoints included treatment-emergent adverse events (TEAEs) and serious adverse events (SAEs). Investigator-reported infusion-related reactions (IRRs) were also summarized. For exploratory analyses, tumor tissue was assessed by immunohistochemistry for expression of PDGFR-α and PDGFR-β to test for correlation to OS and PFS.

### 2.5. PRO Assessment

The European Organization for Research and Treatment of Cancer (EORTC) quality of life questionnaire (QLQ-C30) [20] and the modified Brief Pain Inventory-short form (mBPI-sf) [21,22] were included in the Phase 2 portion of the trial. Each instrument was completed on Day 1 of every treatment cycle in the clinic prior to receiving the study drug. A clinically meaningful difference was defined a priori as a change of 10 points or more from each participant’s own baseline score on QLQ-C30 subscales [23] and as an increase of ≥2 points from baseline on mBPI-sf [21,22]. Time to first worsening (defined as the time from randomization to the first observations of worsening) is described using the method of Kaplan–Meier and analyses were made between the two arms by a log-rank test.

### 2.6. Pharmacokinetics

The olaratumab data were analyzed using an established population PK model [24] using NONMEM Version 7.4.2 (ICON Plc, Gaithersburg, MD, USA). Docetaxel, gemcitabine, and its metabolite dFdU data were analyzed by standard non-compartmental analysis using Phoenix^®^ WinNonlin^®^ 8.1 (Pharsight, A Certara Company, Princeton, NJ, USA) or data were compared using descriptive statistics.

### 2.7. Statistical Analysis

All preplanned analyses were performed separately for O-naïve and O-pretreated cohorts. The efficacy analyses were performed in the randomization patient population (intention-to-treat (ITT) population). OS and PFS were compared between treatment arms based on a stratified log-rank test, stratified by three randomization strata; that is, number of prior systemic therapies for advanced/metastatic disease, histological tumor type, and ECOG PS. OS, PFS, and PROs were analyzed using the Kaplan–Meier (KM) method; median and exact 95% confidence intervals (CIs) were estimated. The hazard ratio (HR) was estimated using a Cox regression model stratified by the three randomization strata. ORR and DCR were summarized and included exact 95% CIs. The safety analyses were performed in the population of patients who received at least one dose of study treatment (safety population). Safety data, such as TEAEs and deaths on study therapy, were summarized as the percent of patients with one/more events. The primary analysis was OS in the O-naïve cohort, conducted with a 2-sided 0.20 alpha level testing. All other comparative analyses were considered exploratory if the primary analysis was found not to be statistically significant. Analyses were conducted using SAS software Enterprise Guide 7.15 (SAS Institute Inc., Cary, NC, USA).

## 3. Results

### 3.1. Descriptive Analysis

In total, 256 patients were enrolled in the Phase 2 study: 167 in the O-naïve cohort and 89 in the O-pretreated cohort (Appendix A). Baseline patient characteristics, disease extent, and prior therapy characteristics are summarized (Table 1) and were found to be well balanced between the investigational and control arms of both cohorts, with only minor differences in baseline patient characteristics.

### 3.2. Efficacy

The primary endpoint of this study was not reached; median OS in the ITT population in the O-naïve cohort was not significantly longer for the investigational arm (16.8 months) than for the control arm (18.0 months) (HR = 0.95; 95% CI = 0.64−1.40; *p* = 0.78) (Figure 1C). As the primary analysis was found not to be statistically significant, all other comparative analyses were considered exploratory. In the O-pretreated cohort, the median OS was 19.8 months and 17.3 months for the investigational and control arms, respectively (HR = 0.67; 95% CI = 0.39−1.16; *p* = 0.15) (Figure 1D).

The investigational arm reported a longer median PFS (7.6 months) than for the control arm (4.4 months) for O-naïve patients (HR = 0.69; 95% CI = 0.48−1.01; *p* = 0.06) (Figure 1A). Median PFS in the O-pretreated cohort was 5.5 months in the investigational arm and 4.2 months in the control arm (HR = 0.83; 95% CI = 0.49−1.40; *p* = 0.48) (Figure 1B).

ORRs in the O-naïve cohort were 32.1% and 23.3% (*p* = 0.19) and in the O-pretreated cohort 30.4% and 14.0% in the investigational arm and control arm (*p* = 0.06), respectively. DCRs were 74.1% in the investigational arm and 72.1% in the control arm (*p* = 0.77) for the O-naïve cohort and were 67.4% and 62.8% (*p* = 0.65) for the O-pretreated cohort, respectively (Table 2). Median DoRs (unconfirmed) in the O-naïve cohort were 5.6 months (95% CI = 4.0−8.1) and 9.9 months (95% CI = 5.2–NR) in the investigational arm and control arm, and in the O-pretreated cohort they were 12.4 months (95% CI = 3.0−15.9) and 8.3 months (95% CI = 1.4–NR), respectively (Table 2).

Separately, a prespecified subgroup analysis of OS in different STS histological subtypes (LMS and non-LMS) was undertaken (Appendix A). In both cohorts, there was no statistically significant OS benefit for the investigational arm in the LMS/non-LMS group (Appendix A). In a retrospective exploratory analysis of the major histological tumor types (liposarcoma, LMS, undifferentiated pleomorphic sarcoma, and others) pooled between O-naïve and O-pretreated, no clinically significant differences in OS or PFS were observed between treatment arms (Appendix A). However, there was evidence of substantial histologic heterogeneity, which underlies a fundamental methodological challenge of trying to study STS.

### 3.3. Safety

A total of 255 patients of which 126 patients were from the investigational arm and 129 patients were from the control arm were included in the safety analysis.

An overview of safety by cohort is presented in Table 3.

For the O-naïve cohort, a similar safety profile was observed across both the investigational and control arms. The most frequent adverse events were fatigue (investigational = 75.3% vs. control = 65.1%), anemia (investigational = 56.8% vs. control=55.8%), musculoskeletal pain (investigational = 50.6% vs. control = 44.2%), and nausea (investigational = 46.9% vs. control = 46.5%) (Table 4).

Grade ≥ 3 TEAEs were reported by 84.0% of patients in the investigational arm and 73.3% in the control arm. The most frequently reported grade ≥ 3 TEAEs were anemia (investigational = 22.2% vs. control = 20.9%), neutropenia (investigational = 33.3% vs. control = 25.6%), and thrombocytopenia (investigational=22.2% vs. control = 17.4%). In the O-naïve cohort, a higher proportion of patients in the investigational arm experienced IRRs compared to the control arm (any grade, 18.5% vs. 5.8%; grade ≥3, 3.7% vs. 0.0%). Two deaths due to an adverse event (skull fracture and myocardial infarction) were reported in the control arm of the O-naïve cohort while on study treatment/within 30 days of discontinuation that were not related to study treatment.

The O-pretreated cohort had a generally consistent safety profile across both the investigational and control arms, except for a more frequent occurrence of grade ≥3 TEAEs of anemia (investigational = 26.7% vs. control = 14.0%), thrombocytopenia (investigational = 22.2% vs. control = 11.6%), neutropenia (investigational = 44.4% vs. control = 14.0%), and leukopenia (investigational = 17.8% vs. control = 2.3%) in the investigational arm. In the O-pretreated cohort, a similar proportion of patients experienced any-grade IRRs in both the control and investigational arm (any grade, 11.1% vs. 9.3%; grade ≥3, 0.0% vs. 0.0%). The O-pretreated cohort reported one death in the investigative arm (hepatic failure) and one death in the control arm (respiratory failure) due to an adverse event while on study treatment or within 30 days of discontinuation that were related to study treatment.

### 3.4. Patient-Reported Outcomes

All patients in the O-naïve cohort completed a baseline C30 questionnaire; 82.7% in the O+G+D arm and 82.6% in the PBO+G+D arm completed a baseline mBPI-sf questionnaire. In the O-pretreated cohort, 97.8% in the O+G+D arm and 100% in the PBO+G+D arm completed a baseline C30 questionnaire; and 89.1% and 79.1%, respectively, completed a mBPI-sf baseline assessment. Baseline scores within groups were similar. There were no differences in time to worsening for the global health status/quality of life subscale or mBPI-sf time to first worsening results between study cohorts (Appendix A). 

### 3.5. Pharmacokinetics

Olaratumab PK data from 178 patients in Phase 1b/2 were combined with PK data from a previous population PK analysis. The systemic clearance was 0.0186 L/h in the current analysis versus 0.0193 L/h in the previous analysis. Additionally, the volume of distribution was 5.05 L in the current analysis versus 5.62 L in the previous analysis.

Docetaxel PK data from 268 patients were analyzed. All data were dose normalized to 75 mg/m^2^. In Phase 1b, the geometric mean AUC (0–∞) for docetaxel was 3470 ng ·h/mL and 3200 ng·h/mL when given with olaratumab (15 and 20 mg/kg) and 900 mg/m^2^ gemcitabine, respectively. In Phase 1b/2, the C_max_ for docetaxel ranged from 817 ng/mL to 1260 ng/mL when given with gemcitabine and olaratumab or placebo. 

Gemcitabine and its metabolite dFdU (2′,2′-difluoro-2′-deoxyuridine) PK data were collected from 272 patients from Phase 1b/2. All data were dose normalized to 900 mg/m^2^. The geometric mean C_max_ for gemcitabine ranged from 2.65 μg/mL to 4.00 μg/mL following a 90 min infusion of gemcitabine with olaratumab (15 or 20 mg/kg) or placebo with docetaxel (75 mg/m^2^). The geometric mean C_max_ for dFdU ranged from 26.8 μg/mL to 29.4 μg/mL.

### 3.6. PDGFR-α and PDGFR-β Expression

Finally, an exploratory analysis was conducted to see if PDGFR-α and PDGFR-β could be used as a predictive biomarker in patients with advanced STS. Overall, we found that a higher proportion of patients in both cohorts presented with PDGFR-α- and PDGFR-β-positive tumors than negative. Exploratory analyses of OS and PFS by PDGFR-α and PDGFR-β tumor status found no association between PDGFR-α or PDGFR-β expression and olaratumab response in the O-naïve or O-pretreated cohorts (Appendix A).

## 4. Discussion

The primary endpoint of the Phase 2 ANNOUNCE 2 trial was not met; there was no statistically significant difference in OS between the two arms in the O-naïve cohort. Based on the statistical plan, all other outcomes in this study are considered exploratory, and therefore we cannot claim any statistically significant difference. While OS was not statistically significant, the combination of O+G+D showed favorable results for other efficacy endpoints in the O-naïve cohort. For instance, the investigational arm had a longer median PFS compared to the control arm (HR = 0.69; *p* = 0.06) and a higher ORR (32.1% vs. 23.3%; *p* = 0.19). The potential olaratumab benefit in PFS in the O-naïve population is possibly driven by delaying the appearance of new lesions (Appendix A).

In the O-pretreated cohort, the trend in the efficacy outcomes suggest the hypothesis that olaratumab may have more impact with longer-term continuous usage. Following randomization, patients in the investigational arm had numerically better outcomes than those in the control arm of the O-pretreated cohort. It is possible that some selection bias may have existed in the enrollment of the O-pretreated patients. For example, investigators may have had some tendency to randomize only patients who were perceived to have done well with olaratumab in 1L therapy.

Some numerical differences in OS HRs by histological tumor type were observed. For example, in the liposarcoma subgroup an estimated OS HR of 1.63 was observed; in the earlier ANNOUNCE trial of olaratumab in combination with doxorubicin, an OS HR of 1.29 had been reported in liposarcoma patients [18]. These results suggest that the effect of olaratumab may not be uniform within non-LMS histologic types, many of which are quite rare. Heterogeneity of efficacy, especially within non-LMS histologic types, may ultimately be the reason for an observed heterogeneity of results from different clinical trials of olaratumab. Furthermore, the lack of OS benefit in liposarcoma patients is consistent with the other two olaratumab STS clinical trial results observed for liposarcoma [18,19].

Overall, a tolerable safety profile was observed across treatment arms for both cohorts, with no new safety signals observed.

PRO data show comparable outcomes in QLQ-C30 global health status/quality of life subscale and mBPI-sf outcomes between the interventional and control arms for both cohorts. However, this was not the case for all subscales.

The PK analysis results for olaratumab, from both the Phase 1b and 2 portions of this study, aligned with previous clinical studies. Similarly, the plasma PK of gemcitabine, its metabolite dFdU, and docetaxel were similar across cohorts and comparable to historic data indicating that olaratumab did not influence the PK of the other drugs and metabolites. Therefore, it is unlikely that variability in PK, for any of the studied drugs, contributed to a lack of improved outcomes with the interventional arm in this study.

PDGFR expression was shown to have no correlation with clinical outcome, a finding that was also observed in ANNOUNCE Phases 2 and 3 trials. Although exploratory in nature, these findings underscore the clinical need for a greater knowledge of STS biology to develop effective treatments.

Trials of olaratumab in STS clearly illustrate the challenges of drug development in these rare and biologically heterogenous cancers. Despite the promising OS benefit seen for the combination of doxorubicin and olaratumab in the randomized Phase 2 trial, the subsequent Phase 3 trial did not meet the OS primary endpoint in the entire STS cohort and in the LMS cohort. The rationale for two co-primary endpoints (OS in overall STS and LMS cohorts) in the Phase 3 trial was to account for the heterogeneity of a composite endpoint such as OS in an all-comer STS trial. However, there is significant heterogeneity even within a specific subgroup such as LMS, and previous studies have shown that there are distinct anatomic and molecular variants of LMS [25,26]. Consequently, future trials of novel agents in STS should incorporate the evaluation of putative biomarkers into trial design as well as focus on drug mechanism of action.

However, the current trial gives further evidence that the outcome of patients with advanced/metastatic STS is improving. The median OS for patients diagnosed with advanced/metastatic STS was previously cited as approximately 12 months [25]. In this trial, the median OS of patients treated beyond 1L therapy ranged from 16.8–19.8 months which is comparable to, or longer than, other clinical trials of gemcitabine–docetaxel and doxorubicin in STS [5,27,28]. Furthermore, several agents have been approved for pretreated STS including pazopanib, trabectedin, and eribulin.

## 5. Conclusions

This clinical trial failed to reach its primary endpoint, therefore no other outcomes in this study can be considered statistically significant. However, the combination of O+G+D showed a possible favorable trend in OS in the O-pretreated cohort and other efficacy outcomes in both cohorts. To date, no clear correlation has been found between PDGFR-α expression and olaratumab efficacy. While this may be an indication that the target is not clinically relevant, it may also be an indication that the role of PDGFR-α expression is more complex and heterogenous than can be detected by IHC. This clinical trial demonstrates the feasibility of testing new agents in rare diseases like sarcoma, with the successful accrual of ˃250 patients at multiple collaborating cancer centers in the USA and Europe.

## Figures and Tables

**Figure 1 cancers-15-04871-f001:**
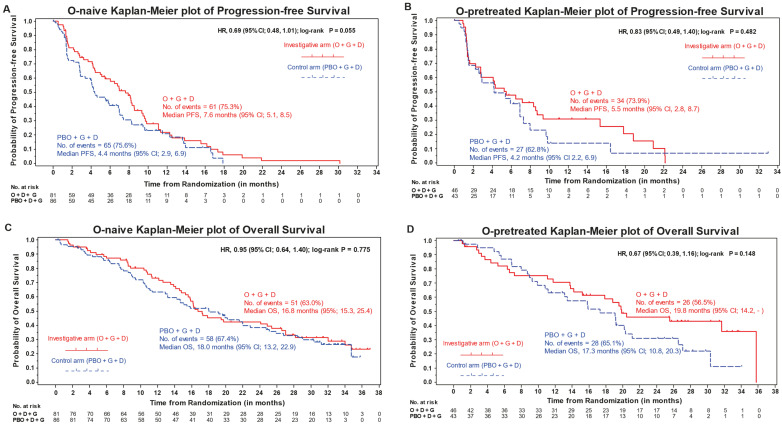
KM plots of OS and PFS for the O-naïve and O-pretreated cohorts (ITT population). (**A**,**B**), KM plots of PFS for the O-naïve and O-pretreated cohorts, respectively. (**C**,**D**), KM plots of OS for the O-naïve and O-pretreated co-horts, respectively. Abbreviations: CI = confidence interval; D = docetaxel; G = gemcitabine; HR = hazard ratio; ITT = intent to treat; KM = Kaplan–Meier; O = olaratumab; OS = overall survival; PBO = placebo; PFS = progression-free survival.

**Table 1 cancers-15-04871-t001:** Demographics and baseline clinical characteristics for the ITT population.

n (%)	Olaratumab-Naïve		Olaratumab-Pretreated	
Cohort	Investigational Arm (O + G + D) N = 81	Control Arm (PBO + G + D) N = 86	Investigational Arm (O + G + D) N = 46	Control Arm (PBO + G + D) N = 43
Sex, Female	48 (59.3)	58 (67.4)	28 (60.9)	28 (65.1)
Age, years, mean (SD)	53.4 (13.4)	53.7 (13.4)	60.5 (11.6)	57.1 (12.3)
<65	66 (81.5)	66 (76.7)	27 (58.7)	30 (69.8)
≥65	15 (18.5)	20 (23.3)	19 (41.3)	13 (30.2)
Race, n (%)				
Asian	3 (3.7)	2 (2.3)	2 (4.3)	2 (4.7)
Black or African American	1 (1.2)	1 (1.2)	8 (17.4)	2 (4.7)
Native Hawaiian or other Pacific Islander	0 (0.0)	0 (0.0)	1 (2.2)	0 (0.0)
White	61 (75.3)	70 (81.4)	30 (65.2)	34 (79.1)
Other	16 (19.8)	13 (15.1)	5 (10.9)	5 (11.6)
ECOG PS ^a^				
0	43 (53.8)	49 (57.6)	26 (57.8)	22 (51.2)
1	37 (46.3)	36 (42.4)	19 (42.2)	21 (48.8)
Pathological diagnosis type				
Leiomyosarcoma	37 (45.7)	38 (44.2)	21 (45.7)	20 (46.5)
Non-leiomyosarcoma	44 (54.3)	48 (55.8)	25 (54.3)	23 (53.5)
Study entry: Histopathological grade				
G1	2 (2.5)	2 (2.3)	0 (0.0)	2 (4.7)
G2	1 (1.2)	3 (3.5)	3 (6.5)	2 (4.7)
G3	15 (18.5)	22 (25.6)	17 (37.0)	11 (25.6)
Other/unknown	63 (77.8)	59 (68.6)	26 (56.5)	28 (65.1)
Prior anticancer therapies				
Prior radiotherapy	35 (43.2)	44 (51.2)	21 (45.7)	21 (48.8)
Prior surgical procedure	70 (86.4)	80 (93.0)	36 (78.3)	39 (90.7)
Systemic therapy	57 (70.4)	65 (75.6)	46 (100)	43 (100)
Number of regimens				
1	41 (50.6)	41 (47.7)	32 (69.6)	32 (74.4)
2	11 (13.6)	12 (14.0)	11 (23.9)	11 (25.6)
3	0 (0.0)	1 (1.2)	2 (4.3)	0 (0.0)

Abbreviations: ECOG PS = Eastern Cooperative Oncology Group performance status; G = grade; ITT = intent to treat; N = number of patients in population; n = number of patients in the specified category; PBO = placebo; SD = standard deviation. ^a^ Number of patients with non-missing data, used as the denominator.

**Table 2 cancers-15-04871-t002:** Summary of antitumor activity.

n (%)	Olaratumab-Naïve			Olaratumab-Pretreated		
Arm	Investigational Arm (O + G + D) N = 81	Control Arm (PBO + G + D) N = 86	HR (95% CI) [*p*-Value] ^c^	Investigational Arm (O + G + D) N = 46	Control Arm (PBO + G + D) N = 43	HR (95% CI) [*p*-Value] ^c, d^
Best overall response ^a^						
Complete response (CR)	1 (1.2)	1 (1.2)		0 (0.0)	0 (0.0)	
Partial response (PR)	25 (30.9)	19 (22.1)		14 (30.4)	6 (14.0)	
Stable disease (SD)	34 (42.0)	42 (48.8)		17 (37.0)	21 (48.8)	
Progressive disease (PD)	16 (19.8)	21 (24.4)		11 (23.9)	11 (25.6)	
Non-evaluable	5 (6.2)	3 (3.5)		4 (8.7)	5 (11.6)	
ORR, n (CR+PR; 95% Cl ^b^)	26 (32.1; 22.2–43.4)	20 (23.3; 14.8–33.6)	0.19	14 (30.4; 17.7–45.8)	6 (14.0; 5.3–27.9)	0.06
DCR, n (CR+PR+SD; 95% CI ^b^)	60 (74.1; 63.1–83.2)	62 (72.1; 61.4–81.2)	0.77	31 (67.4; 52.0–80.5)	27 (62.8; 46.7–77.0)	0.65
Median DoR (95% CI ^c^)	5.6 (4.0–8.1)	9.9 (5.2–)		12.4 (3.0–15.9)	8.3 (1.4–)	
PFS, median mos (95% CI)	7.6 (5.1–8.5)	4.4 (2.9–6.9)	0.7 (0.48–1.01) [0.06]	5.5 (2.76–8.71)	4.2 (2.2–6.9)	0.8 (0.5–1.4) [0.48]
OS, median mos (95% CI)	16.8 (15.3–25.4)	18.0 (13.2–22.9)	0.9 (0.6–1.4) [0.78]	19.8 (14.19–)	17.3 (10.8–20.3)	0.7 (0.4–1.2) [0.15]

Abbreviations: CI = confidence interval; Control = placebo + gemcitabine + docetaxel; D = docetaxel; DCR = disease control rate; G = gemcitabine; HR = hazard ratio; N = number of subjects in population; n = number of patients in the specified category; O = olaratumab; ORR = overall response rate; OS = overall survival; PFS = progression-free survival. ^a^ Response criteria used were RECIST 1.1. ^b^ Confidence intervals are based on the exact method. ^c^ Confidence intervals were estimated using Greenwood method. ^d^ Log-rank *p*-value (2-sided); *p*-value is calculated by exact Mantel–Haenszel test stratified by interactive web-response system stratification factors.

**Table 3 cancers-15-04871-t003:** Summary of safety.

Number of Patients ^a^ n (%)	Olaratumab-Naïve		Olaratumab-Pretreated	
Cohort	Investigational Arm (O+G+D) N = 81	Control Arm (PBO+G+D) N = 86	Investigational Arm (O+G+D) N = 45	Control Arm (PBO+G+D) N = 43
Exposure to olaratumab or placebo	81 (100.0)	86 (100.0)	45 (100.0)	43 (100.0)
Duration of treatment, weeks, median (range)	19.0 (3.0–134.9)	18.5 (2.9–84.1)	18.0 (3.0–99.0)	12.0 (3.0–150.0)
Cycles received per patient ^a^, median (range)	6.0 (1.0–44.0)	5.5 (1.0–27.0)	6.0 (1.0–32.0)	4.0 (1.0–48.0)
Exposure to gemcitabine	79 (97.5)	86 (100.0)	45 (100.0)	43 (100.0)
Duration of treatment, weeks, median (range)	18.3 (3.0–94.1)	15.0 (2.9–77.3)	18.0 (3.0–99.0)	12.0 (3.0–150.0)
Cycles received per patient ^a^, median (range)	6.0 (1.0–27.0)	5.0 (1.0–24.0)	6.0 (1.0–32.0)	4.0 (1.0–48.0)
Exposure to docetaxel	76 (93.8)	83 (96.5)	41 (91.1)	41 (95.3)
Duration of treatment, weeks, median (range)	15.8 (2.0–94.0)	12.0 (2.0–59.9)	17.0 (2.0–149.0)	11.0 (2.0–149.0)
Cycles received per patient ^a^, median (range)	5.0 (1.0–29.0)	4.0 (1.0–19.0)	5.0 (1.0–32.0)	4.0 (1.0–46.0)
Dose adjustments				
Patients with at least one dose adjustment	73 (90.1)	67 (77.9)	38 (84.4)	30 (69.8)
Patients with at least one dose reduction	51 (63.0)	39 (45.3)	28 (62.2)	18 (41.9)
Adverse events in the safety population				
Patients with ≥1 TEAE	81 (100.0)	83 (96.5)	45 (100.0)	43 (100.0)
Related to study treatment ^b^	77 (95.1)	78 (90.7)	43 (95.6)	40 (93.0)
Patients with ≥1 grade ≥3 TEAE	68 (84.0)	63 (73.3)	38 (84.4)	35 (81.4)
Related to study treatment ^b^	59 (72.8)	51 (59.3)	32 (71.1)	29 (67.4)
Patients with ≥1 SAE	44 (54.3)	38 (44.2)	21 (46.7)	23 (53.5)
Related to study treatment ^b^	33 (40.7)	24 (27.9)	13 (28.9)	17 (39.5)
Patients who discontinued study treatment due to AE	15 (18.5)	13 (15.1)	6 (13.3)	10 (23.3)
Related to study treatment ^b^	12 (14.8)	8 (9.3)	5 (11.1)	8 (18.6)
Patients who died due to AE on study treatment or within 30 days of discontinuation ^c^	0 (0.0)	2 (2.3)	1 (2.2)	1 (2.3)
Related to study treatment ^b^	0 (0.0)	0 (0.0)	1 (2.2)	1 (2.3)

Abbreviations: AE = adverse event; N = number of treated patients; n = number of patients in category; SAE = serious adverse event; TEAE = treatment-emergent adverse event. ^a^ Subjects may be counted in more than one category; ^b^ events that were considered related to study treatment as judged by the investigator; ^c^ in the control arm of the O-naïve cohort, two deaths were reported: skull fracture and myocardial infraction. In the O-pretreated cohort, two deaths were reported: hepatic failure (investigative arm) and respiratory failure (control arm).

**Table 4 cancers-15-04871-t004:** Treatment-emergent adverse events (all-causality) occurring in ≥25% of patients who received ≥1 dose of study treatment (in either cohort, any grade).

Number of Patients n (%)	Olaratumab-Naïve				Number of Patients n (%)	Olaratumab-Pretreated			
Cohort	Investigational Arm (O+G+D) N = 81		Control Arm (PBO+G+D) N = 86			Investigational Arm (O+G+D) N = 45		Control Arm (PBO+G+D) N = 43	
MedDRA Preferred Term	All Grades ^a^	≥ Grade 3	All Grades ^a^	≥ Grade 3	MedDRA Preferred Term	All Grades ^a^	≥ Grade 3	All Grades ^a^	≥ Grade 3
Fatigue ^b^	61 (75.3)	12 (14.8)	56 (65.1)	7 (8.1)	Anemia ^b^	34 (75.6)	12 (26.7)	20 (46.5)	6 (14.0)
Musculoskeletal pain ^b^	41 (50.6)	3 (3.7)	38 (44.2)	2 (2.3)	Fatigue ^b^	34 (75.6)	7 (15.6)	25 (58.1)	4 (9.3)
Anemia ^b^	46 (56.8)	18 (22.2)	48 (55.8)	18 (20.9)	Musculoskeletal pain ^b^	31 (68.9)	2 (4.4)	22 (51.2)	4 (9.3)
Nausea	38 (46.9)	2 (2.5)	40 (46.5)	3 (3.5)	Neutropenia ^b^	27 (60.0)	20 (44.4)	8 (18.6)	6 (14.0)
Oedema peripheral	38 (46.9)	3 (3.7)	23 (26.7)	1 (1.2)	Thrombocytopenia ^b^	22 (48.9)	10 (22.2)	13 (30.2)	5 (11.6)
Diarrhea	37 (45.7)	5 (6.2)	31 (36.0)	1 (1.2)	Diarrhea	21 (46.7)	1 (2.2)	14 (32.6)	1 (2.3)
Neutropenia ^b^	34 (42.0)	27 (33.3)	37 (43.0)	22 (25.6)	Oedema peripheral	21 (46.7)	1 (2.2)	14 (32.6)	1(2.3)
Alopecia	29 (35.8)	1 (1.2)	32 (37.2)	0 (0.0)	Nausea	20 (44.4)	0 (0.0)	12 (27.9)	0 (0.0)
Thrombocytopenia ^b^	28 (34.6)	18 (22.2)	26 (30.2)	15 (17.4)	Dyspnea	16 (35.6)	3 (6.7)	13 (30.2)	3 (7.0)
Pyrexia	28 (34.6)	1 (1.2)	28 (32.6)	1 (1.2)	Cough	13 (28.9)	0 (0.0)	9 (20.9)	0 (0.0)
Vomiting	25 (30.9)	1 (1.2)	13 (15.1)	1 (1.2)	Constipation	13 (28.9)	1 (2.2)	12 (27.9)	0 (0.0)
Constipation	21 (25.9)	0 (0.0)	20 (23.3)	0 (0.0)	Dysgeusia	13 (28.9)	0 (0.0)	14 (32.6)	0 (0.0)
Decreased appetite	21 (25.9)	0 (0.0)	15 (17.4)	1 (1.2)	Leukopenia ^b^	13 (28.9)	8 (17.8)	5 (11.6)	1 (2.3)
Dyspnea	21 (25.9)	1 (1.2)	21 (24.4)	4 (4.7)	-	-	-	-	-
AESI					AESI				
Infusion-related reactions ^c^					Infusion-related reactions ^c^				
Investigator reported	15 (18.5)	3 (3.7)	5 (5.8)	0 (0.0)	Investigator reported	2 (4.4)	0 (0.0)	1 (2.3)	0 (0.0)
Algorithm derived	15 (18.5)	3 (3.7)	13 (15.1)	0 (0.0)	Algorithm derived	5 (11.1)	0 (0.0)	4 (9.3)	0 (0.0)

Abbreviations: AESI = adverse event of special interest; MedDRA = Medical Dictionary for Regulatory Activities; ^a^ Common Terminology Criteria for Adverse Events version 4.0 was used to categorize TEAEs. Grades 1–5 were defined as: mild (grade 1), moderate (grade 2), severe or medically significant but not immediately life-threatening (grade 3), life-threatening (grade 4), and death related to TEAE (grade 5). ^b^ The following consolidated terms include terms presented in parentheses: anemia (anemia; hemoglobin decreased, red blood cell count decreased); fatigue (fatigue, asthenia); leukopenia (leukopenia, white blood cell count decreased); musculoskeletal pain (arthralgia, back pain, pain in extremity, muscle spasms, myalgia, bone pain, musculoskeletal chest pain, groin pain, neck pain, flank pain); neutropenia (neutrophil count decreased); and thrombocytopenia (platelet count decreased). ^c^ Only reporting immediate IRRs. N = number of subjects in population; n = number of patients in the specified category.

## Data Availability

Lilly provides access to all individual participant data collected during the trial, after anonymization, with the exception of pharmacokinetic or genetic data. Data are available to request 6 months after the indication studied has been approved in the US and EU and after primary publication acceptance, whichever is later. No expiration date of data requests is currently set once data are made available. Access is provided after a proposal has been approved by an independent review committee identified for this purpose and after receipt of a signed data sharing agreement. Data and documents, including the study protocol, statistical analysis plan, clinical study report, blank or annotated case report forms, will be provided in a secure data sharing environment. For details on submitting a request, see the instructions provided at www.vivli.org.

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
