# Peer review of "Randomized Phase 2 Clinical Trial of Olaratumab in Combination with Gemcitabine and Docetaxel in Advanced Soft Tissue Sarcomas"

_cancers, 2023, doi:10.3390/cancers15194871_

Round 1

Reviewer 1 Report

Attia and colleagues report the results of a phase 2 randomized clinical trial evaluating olaratumab (O) in combination with gemcitabine (G) and docetaxel (D) in advanced soft tissue sarcomas. The study included two cohorts: an O-naive group (N=167) and and O-pretreated group (N=89). 

No difference in the primary endpoint, which was a difference in overall survival between O+G+D vs G+D in the O-naive group, was observed.  This is similar to a prior ANNOUNCE-2 evaluating olaratumab + doxorubicin in STS, in which a survival difference was also not observed.

The authors concluded that a difference was noted in the O-pretreated group, where the PFS was 5.5mo vs 4.2mo and O.S. was 19.8mo vs 17.3mo. However, these difference in survival do not appear to be significant, nor was the study designed to answer this question. Therefore, the 2nd to last line of the abstract indicating favorable outcomes for O+G+D in the O-pretreated group should be omitted. The results/discussion should also be modified to suggest a possible trend only and that it supports the hypothesis of a favorable outcomes for O+G+D. 

This study was otherwise well done and the manuscript was well written.

Author Response

Thank you for the opportunity to submit a revised draft of our manuscript entitled “Randomized Phase 2 clinical trial of olaratumab in combination with gemcitabine and docetaxel in advanced soft tissue sarcomas” to Cancers. We appreciate the time that the reviewers dedicated to providing feedback and are grateful for their insightful comments. Please find below our point-by-point responses.

REVIEWER 1

Attia and colleagues report the results of a phase 2 randomized clinical trial evaluating olaratumab (O) in combination with gemcitabine (G) and docetaxel (D) in advanced soft tissue sarcomas. The study included two cohorts: an O-naive group (N=167) and and O-pretreated group (N=89). 

No difference in the primary endpoint, which was a difference in overall survival between O+G+D vs G+D in the O-naive group, was observed.  This is similar to a prior ANNOUNCE-2 evaluating olaratumab + doxorubicin in STS, in which a survival difference was also not observed.

The authors concluded that a difference was noted in the O-pretreated group, where the PFS was 5.5mo vs 4.2mo and O.S. was 19.8mo vs 17.3mo. However, these difference in survival do not appear to be significant, nor was the study designed to answer this question. Therefore, the 2nd to last line of the abstract indicating favorable outcomes for O+G+D in the O-pretreated group should be omitted. The results/discussion should also be modified to suggest a possible trend only and that it supports the hypothesis of a favorable outcomes for O+G+D. 

This study was otherwise well done and the manuscript was well written.

Response: We have deleted the sentence as per suggestion of the Reviewer, including that sentence in the Simple Summary. We have stream-lined the corresponding paragraph in the Discussion and modified the text to clarify that outcomes suggest only a possible trend in support of a favorable outcome for O+G+D. We also modified a related sentence in the conclusions.

Reviewer 2 Report

Dear Authors; I found your work title "Randomized Phase 2 clinical trial of olaratumab in combination with gemcitabine and docetaxel in advanced soft tissue sarcomas" an interesting addition to the current literature. It needs some extra work prior to processing it further. Regards. P.S.

[1] Writing:

1-1 Add list of used abbreviation in the work and their expansions at the end of text right before Reference section for reader's access. 

1-2 Discussion Section. It is hard to follow. Break it down to some subsections. Something like this: 4.1 Summary & Contributions; 4.2. Strengths & Limitations; 4.3. Future Work

1-3 Rename Section "3.1.Patients" to "3.1. Descriptive Analysis". 

[2] Statistical

2-1 Statistical Analysis Software: Which software did you use ? R/SAS/STATA/ SPSS/ etc. ? Add and cite its reference in section "2.7. Statistical Analysis".

2-2 Defend/Eplain your choice of taking alpha=0.20 in line 197.

2-3 Why did you not use Cox Proportional Hazard (CPH) Models or Accelerated Failure Time (AFT) Models in your survival analysis ? Defend your abstinence or add the results of one of these models in your section "3.Analysis".

Author Response

Thank you for the opportunity to submit a revised draft of our manuscript entitled “Randomized Phase 2 clinical trial of olaratumab in combination with gemcitabine and docetaxel in advanced soft tissue sarcomas” to Cancers. We appreciate the time that the reviewers dedicated to providing feedback and are grateful for their insightful comments. Please find below our point-by-point responses.

REVIEWER 2

Dear Authors; I found your work title "Randomized Phase 2 clinical trial of olaratumab in combination with gemcitabine and docetaxel in advanced soft tissue sarcomas" an interesting addition to the current literature. It needs some extra work prior to processing it further. Regards. P.S.

[1] Writing:

1-1 Add list of used abbreviation in the work and their expansions at the end of text right before Reference section for reader's access. 

Response: A list of abbreviations and their expansions was created and placed at the end of text before References.

1-2 Discussion Section. It is hard to follow. Break it down to some subsections. Something like this: 4.1 Summary & Contributions; 4.2. Strengths & Limitations; 4.3. Future Work

Response: We modified the order of the paragraphs to make it easier to follow. We present the findings of the study, followed by limitations and future work.

1-3 Rename Section "3.1.Patients" to "3.1. Descriptive Analysis". 

Response: Section 3.1 was renamed "Descriptive Analysis".

[2] Statistical

2-1 Statistical Analysis Software: Which software did you use ? R/SAS/STATA/ SPSS/ etc. ? Add and cite its reference in section "2.7. Statistical Analysis".

Response: It was an oversight that we did not include this in the Methods Section, and so we added that analyses were conducted using SAS Software (SAS Institute Inc., Cary NC) in the Statistical Analysis section.

2-2 Defend/Eplain your choice of taking alpha=0.20 in line 197.

Response: Applying a 2-sided alpha level of 0.20 is a commonly used method in phase-2 trials, where the goal is to use the most realistic assumptions and test clinically relevant hypotheses, while at the same time acknowledging that it is not a confirmatory phase-3 trial requiring the more rigorous scientific and regulatory standard of a 0.05 alpha level (i.e. We intended to test the same hypotheses we would in a phase-3 trial, and as such the only thing that distinguishes this as a phase-2 study design is the use of a less rigorous, non-confirmatory 0.20 alpha level).

2-3 Why did you not use Cox Proportional Hazard (CPH) Models or Accelerated Failure Time (AFT) Models in your survival analysis ? Defend your abstinence or add the results of one of these models in your section "3.Analysis".

Response: All of the hazard ratio estimates reported in Section 3 are from Cox proportional hazards models. So Cox modelling was extensively applied. It is an oversight that we did not make this clear in the Methods Section, and so the Methods section was updated to identify this fact. The section reads as follows: “OS, PFS and PROs were analyzed using Kaplan-Meier (KM) method; median and exact 95% confidence intervals (CI) were estimated. The hazard ratio (HR) was estimated using a Cox regression model stratified by the three randomization strata.

Reviewer 3 Report

This manuscript describes a randomized placebo-controlled phase II study of olaratumab (O) in advanced soft tissue sarcoma (STS).

The manuscript is well written, and of importance both to those who care for STS patients as well as those who plan clinical trials in rare diseases.

The main result of this trial, in my opinion, is the proof of feasibility of such studies! Advanced STS are a very heterogeneous group of sarcomas, and a very rare disease, thus planning and performing clinical trials is challenging.

The authors, however, have mastered this challenge. They present the results of the trial here in an appropriate manner. The trial design is clearly described, methods are appropriate, and results are presented clearly.

There is one minor point from my side: The authors found no benefit of the investigational drug (O) in drug-naïve patients, and as this was the main goal of the trial, all other results are to be interpreted with caution.

The authors generally adhere to this and state so themselves, but still speculate, that patients who were not O-naïve might have benefited from (further) medication with this substance. I find this a bit challenging, and as the authors themselves state in the discussion, a selection bias might have contributed to this finding (with patients previously experiencing some response from the investigational drug being more likely to be included in a study investigating this drug).

Apart from this, this is an important study. It proves that there likely is no major benefit from adding O to a standard chemotherapy backbone in STS (and negative results need to be published!), and moreover the authors have proven that such studies can (and should) be undertaken even in very rare diseases.

Thus, I strongly recoment publishing these results.

Author Response

Thank you for the opportunity to submit a revised draft of our manuscript entitled “Randomized Phase 2 clinical trial of olaratumab in combination with gemcitabine and docetaxel in advanced soft tissue sarcomas” to Cancers. We appreciate the time that the reviewers dedicated to providing feedback and are grateful for their insightful comments. Please find below our point-by-point responses.

REVIEWER 3

This manuscript describes a randomized placebo-controlled phase II study of olaratumab (O) in advanced soft tissue sarcoma (STS).

The manuscript is well written, and of importance both to those who care for STS patients as well as those who plan clinical trials in rare diseases.

The main result of this trial, in my opinion, is the proof of feasibility of such studies! Advanced STS are a very heterogeneous group of sarcomas, and a very rare disease, thus planning and performing clinical trials is challenging.

The authors, however, have mastered this challenge. They present the results of the trial here in an appropriate manner. The trial design is clearly described, methods are appropriate, and results are presented clearly.

There is one minor point from my side: The authors found no benefit of the investigational drug (O) in drug-naïve patients, and as this was the main goal of the trial, all other results are to be interpreted with caution.

The authors generally adhere to this and state so themselves, but still speculate, that patients who were not O-naïve might have benefited from (further) medication with this substance. I find this a bit challenging, and as the authors themselves state in the discussion, a selection bias might have contributed to this finding (with patients previously experiencing some response from the investigational drug being more likely to be included in a study investigating this drug).

Apart from this, this is an important study. It proves that there likely is no major benefit from adding O to a standard chemotherapy backbone in STS (and negative results need to be published!), and moreover the authors have proven that such studies can (and should) be undertaken even in very rare diseases.

Thus, I strongly recoment publishing these results.

Response: We have edited that paragraph to tone down any claim regarding the benefit of olaratumab in O-pretreted cohort, we have streamlined that paragraph and clarified that it is only an observed trend.

Round 2

Reviewer 2 Report

Dear Authors; most of my comments were addressed satisfactorily. Regards.